# Interaction of Flupyradifurone and Deltamethrin, Two Pesticides Commonly Used for Plant Pest Control, in Honeybees

**DOI:** 10.3390/ani14060851

**Published:** 2024-03-10

**Authors:** Roberto Bava, Carmine Lupia, Fabio Castagna, Stefano Ruga, Saverio Nucera, Cristina Carresi, Rosamaria Caminiti, Rosa Maria Bulotta, Clara Naccari, Domenico Britti, Ernesto Palma

**Affiliations:** 1Department of Health Sciences, University of Catanzaro Magna Græcia, 88100 Catanzaro, Italy; roberto.bava@unicz.it (R.B.); studiolupiacarmine@libero.it (C.L.); rugast1@gmail.com (S.R.); saverio.nucera@hotmail.it (S.N.); carresi@unicz.it (C.C.); rosamariacaminiti4@gmail.com (R.C.); rosamaria.bulotta@gmail.com (R.M.B.); c.naccari@unicz.it (C.N.); britti@unicz.it (D.B.); palma@unicz.it (E.P.); 2Mediterranean Ethnobotanical Conservatory, Sersale (CZ), 88054 Catanzaro, Italy; 3Center for Pharmacological Research, Food Safety, High Tech and Health (IRC-FSH), University of Catanzaro Magna Græcia, 88100 Catanzaro, Italy

**Keywords:** honeybee (*Apis mellifera*), plants, deltamethrin, flupyradifurone, pesticides, drug interaction, antagonism, beekeeping, animal health and welfare

## Abstract

**Simple Summary:**

The exposure of pollinating insects to pesticides is a common phenomenon in ecosystems. Exposure to toxic doses of two or more compounds can lead to additive, synergistic, or antagonistic effects. These drug interactions are often unknown but may amplify or reduce the toxic effect on target and non-target organisms. In the following study, we evaluated whether the combination of two pesticides commonly used in agriculture, deltamethrin and flupyradifurone, was more toxic to honeybees than a single exposure to each pesticide. Although both DMT and FLU were found to be toxic to honeybees, in no case did their combination lead to increased mortality, and, in some cases, it even produced a higher survival rate.

**Abstract:**

Nowadays, old-generation pesticides are released into ecosystems alongside new formulations, giving rise to pharmacological interactions (additive, synergistic, and antagonistic effects). The aim of this study was to evaluate the impact that simultaneous exposure to DMT and FLU doses has on bee health. Groups of twenty honeybees were housed in cages to compose six macro-groups. One group consisted of experimental replicates treated orally with a toxic dose of deltamenthrin (DMT 21.6 mg/L); two other groups were subjected to the oral administration of two toxic doses of flupyradifurone (FLU 50 mg/L and FLU 100 mg/L); and two other groups were intoxicated with a combination of the two pesticides (DMT 21.6 + FLU 50 and DMT 21.6 + FLU 100). The consequences of the pesticides’ interactions were highlighted by measuring and comparing data on survival, food consumption, and abnormal behavior. Generally speaking, antagonism between the two pesticides has been demonstrated. The bees were able to survive for up to three days at the lowest dosage of FLU (50 mg/L), with 46% of the subjects still alive; however, the maximum dose (100 mg/L) caused all treated animals to die as early as the second day. When DMT and FLU 50 were administered together, the group that received DMT alone had a lower survival rate. When comparing the survival rates produced by the DMT and FLU 50 combination to those of the group receiving FLU 50 alone, the same was clearly visible. While there was no statistically significant improvement observed when the survival indices of the DMT and FLU 100 combination were compared to those of the group intoxicated with DMT alone, an improvement in survival indices was observed when these were compared with the group intoxicated with FLU 100 alone.

## 1. Introduction

More than 800 pesticides are used to produce our food [1]. Their use results in enormous benefits in facilitating the production of agricultural products [2]. Despite these benefits, there are negative effects related to the contamination of the environment (contamination of water, soil, air, etc.) and indirectly related to the health of humans and other living beings [2]. In addition, the massive use of pesticides causes resistance phenomena in treated pest populations [3]. For these reasons, there is an ongoing search for new active ingredients and combinations thereof that are environmentally friendly and less prone to the development of drug resistance [4].

Flupyradifurone (FLU) is one of the latest insecticides to have been introduced to the market [5]. FLU can be administered by foliar, soil, or seed application to control hemipterans with stinging or sucking mouthparts that damage crops, such as aphids, aleurodids, and leafhoppers [5]. It acts systemically and spreads via xylem from roots to apices, just like neonicotinoids. The active component disperses uniformly throughout the tissues of the treated botanical species as early as 24 h after application, demonstrating how quickly this translocation occurs [5]. As with other pesticides, compound interaction studies have also been undertaken for FLU.

For the use of pesticides in conjunction with other compounds such as fungicides or insecticides, the Bayer group has filed many patent applications [6,7]; some of them are related to the pesticide FLU [8]. Furthermore, many research groups have carried out research on mixing this pesticide with various fungicides, insecticides, and biological agents [9,10]. This arrangement expands the range of action and application possibilities of pesticide component formulations, in addition to increasing their efficacy. For example, the pyraclostroblin and FLU patent would offer the possibility of controlling animal pests (such as insects, mites, or nematodes) and fungi with an effective mixture [8]. In Le Mauff’s study (2023) [11], the repellent effect of the combination of DEET and FLU on adult females of *Ixodes ricinus* was validated with a new in vitro bioassay. FLU had no repellent effect at any tested concentration but would act as a potentiator of the repellent effect of DEET [11].

Studies have also been conducted on the interaction between FLU and the fungus *Metarhizium brunneum*. At sublethal concentrations, exposure to FLU alone did not affect ant survival, but increased mortality was observed when FLU was combined with *M. brunneum*. FLU is considered bee-safe and can be applied before and throughout the blooming season, being harmless to bees when used in accordance with the prescribed amount on the label [5,12,13]. However, high doses, prolonged usage, and an active ingredient mixture might cause issues for these insects.

Tosi et al. (2019) [13], for instance, provided a summary of the synergistic effects of FLU and other pesticides on honeybees, stating that using propiconazole concurrently with it would result in increased honeybee mortality and aberrant behavior. The impact of FLU on insects was summed up by Siviter et al. (2020) [14]. Its use has led to clear negative effects, even though sublethal effects cannot be seen under typical dosing circumstances. For instance, whether applied at a low or high dosage, the pesticide will lessen the honeybee’s capacity to acquire scents, impairing olfactory learning [15]. This evidence demonstrates the importance of understanding pesticide interactions. The combination of pesticides can determine synergistic and antagonistic effects. A synergistic reaction occurs when two or more chemical compounds combined together produce a greater impact than would occur with the application of each component alone [16]. Because lower application doses than when the chemicals are applied individually are required, less pesticide reaches the environment, which can be advantageous. The combination of the insecticides pyrethrum and piperonyl butoxide is a typical example of a synergistic reaction [17]. If these qualities can be identified in the combination of chemical compounds, the flip side of the coin is revealed in the possible development of antagonistic phenomena and/or resistant strains. The selection of resistant strains, which often have reduced sensitivity at the target site, is an important factor that limits the use of some pesticides in pest control [18,19]. The insect voltage-gated sodium channel was initially affected by DDT and pyrethroids, while acetylcholinesterase (AChE) was affected by methylcarbamates (MCs) and organophosphates (OPs) [20,21]. Instead, the term antagonism refers to a situation where the combined toxicity of the chemicals is less than the sum of their individual effects [16].

The results of the abovementioned studies are even more worrisome when one considers that there are few studies in the literature on the interaction of compounds and that it is very easy for new- and old-generation pesticides to interact with each other in the environment. Thus, the already dramatically known toxic effects peculiar to individual pesticides could be compounded by those brought about by their interaction with other pesticides. The class of pyrethroids includes many insecticides used in agriculture today [22,23]. Professional agriculture, forestry, and hobby farming all make extensive use of deltamethrin, a type II pyrethroid with a broad spectrum of activity [24]. The associated risks to pollinating insects do not exempt its use. Honeybee physiology is disturbed by deltamethrin, which also affects normal dancing and foraging behavior and causes memory problems, hypofertility, hypothermia, and changes in body and intestinal growth [25,26,27,28]. The aim of this study was to determine the consequences that the combination of the new pesticide FLU and the widely used pesticide deltamethrin (DMT) could have on honeybees’ survival and behavior. The same common crops are treated with both FLU and DMT. Since FLU is a relatively new pesticide, few studies have examined the possibility of environmental interactions with other pesticides. Therefore, this study sought to understand whether the combination of DMT and FLU could cause a worsening of honeybee health through a synergistic action of their toxic effects or an improvement through an antagonizing action of one compound on the other.

## 2. Materials and Methods

### 2.1. Honeybee Groups

This research was carried out in the Department of Health Science, University “Magna Græcia” of Catanzaro (Italy), at the Interregional Research Center for Food Safety and Health (IRC-FSH). The experiment was conducted in the summer, as recommended by the guidelines, to avoid the use of honeybees with altered physiology, which occurs in the early spring and late fall [29]. The experimental honeybees were from three hives of *Apis mellifera ligustica*. The colonies were managed according to commonly used beekeeping procedures. Standard inspection procedures were used to ensure that the colonies were healthy before the honeybees were harvested [30,31,32,33]. The honeybees involved in the experiment were collected from a brood honeycomb with honeybees close to emerging. Specifically, the honeycomb was transferred to an incubator at 35 °C and 65-80% relative humidity, and the emerging honeybees were removed after 12 h. This technique facilitates the acquisition of bees of identical age [29]. By brushing the combs, the honeybees were gathered, divided into groups of 20, and placed in cages. The honeybees were placed in the cages using standard procedures, without the use of anesthetics [34].

The cages were equipped with feeders, placed horizontally at the bottom. The feeders were sterile disposable 2.5 mL syringes with closed ends and a cut at the top. Before the start of the experiment, newly hatched honeybees were placed in the cages and fed a sucrose solution (50% sugar in distilled water) from day −1 to day 0. The honeybees were thus able to adapt to the experimental conditions. The experimental groups were kept at 33 ± 2 °C and 70% relative humidity in the dark.

### 2.2. Pesticide Concentration and Administration

Ten replicates of each experimental group were created. Twenty caged honey bees constituted the experimental unit; thus, a total of two hundred honey bees were intoxicated per treatment. One group of honeybees (ten replicates) was treated with DMT at a concentration of 21.6 mg/L dispersed in sucrose solution (50% *w*/*v*). Two other honeybee experimental groups (twenty replicates) were fed two different concentrations of FLU (50 mg/L and 100 mg/L), dispersed in sucrose solution (50% *w*/*v*). In the other groups (20 replicates), the abovementioned concentrations of FLU were combined with the toxic dose of DTM (21.6 mg/L) to verify the effects generated by the pesticide interaction. The ten control (CTRL) groups were treated with sucrose solution only. The concentrations of DTM and FLU used were doses that caused acute intoxication and were chosen by referring to already-published studies [25,35,36,37]. Until administration, the prepared dilutions were kept between 6 and 2 °C and coated in aluminum foil to prevent light deterioration. Every three days, new treatment solutions were prepared. Precipitation never occurred in the feeding solutions. To ensure that the honeybees had enough food throughout the exposure period, all feeders and their contents were changed every 24 h. For a total of 72 h, food intake and survival rates were recorded every 24 h. Additionally, the frequency of aberrant behavior in the honeybees in response to the treatments was monitored at intervals of 1–4, 24, 48, and 72 h after solution administration.

### 2.3. Solution Intake and Honeybee Behavior

The average daily consumption of the solutions was calculated. Solution ingestion was related to the number of surviving honeybees. Evaporation did not need to be taken into account; in fact, in accordance with the “Standard methods for the maintenance of adult *Apis mellifera* in cages under in vitro laboratory conditions” [34], this experimental variable was commonly distributed among all treatment groups. Behavioral disorders were classified and measured according to guidelines published by the OECD [38].

The percentage of honeybees behaving abnormally was monitored at 1, 2, 4, 24, 48, and 72 h after treatment administration. Behaviors of motor incoordination, curved-down abdomen, hyperactivity, decrease in movement activity, and moribundity were considered [13]. Each honeybee in a given cage was observed for six seconds, so for each cage containing twenty honeybees, observation could last up to 120 s.

### 2.4. Statistical Analysis

The statistical analysis was carried out using the GraphPad PRISM software (version 9.0, GraphPad SoftwareInc., La Jolla, CA, USA). The Kaplan–Meier curve was used to determine survival; the Log-rank test was used for multiple comparisons with Bonferroni correction (α = 0.006). A Shapiro–Wilk test for normality was performed. Solution consumption and abnormal bee behavior were analyzed with the Kruskal–Wallis test, followed by the post hoc Dunn’s test.

## 3. Results

### 3.1. Survival Probability

Figure 1 below shows the survival percentage of each group treated during the experiments.

Over the course of the trial’s three days, the CTRL group’s survival rates remained essentially unchanged. After DMT administration, honeybees manifested a significant decline in survival when compared to the CTRL as early as day 1 (*p* < 0.001).

Even the difference in the overall survival rate of the FLU 50 group was statistically lower than that of the CTRL group (*p* < 0.001). FLU 100 was the most toxic, causing a drastic reduction in the survival rate to 18% on day 1, which dropped to 0% on the second day of treatment. When DMT was combined with the lowest concentration of FLU, there was a statistically significant (*p* < 0.001) increase in survival compared to the group treated with DMT alone.

The combination of DMT with FLU 50 also resulted in a statistically significant (*p* < 0.001) improvement in survival compared with the group given the FLU 50 toxic dose alone. When DMT was combined with the more toxic dose of FLU, there was an improvement in survival rate on day 1 compared with the group given the DMT dose alone, but this improvement was lost on subsequent days.

On the other hand, the combination of DMT + FLU 100 produced a statistically significant (*p* < 0.001) improvement in survival rates compared to the group given FLU 100 alone, with the honeybees successfully managing to survive up to the third day of treatment (Figure 1, Table 1).

### 3.2. Solution Intake and Consumption

Figure 2 below shows the data on daily food consumption in the experimental groups.

By dividing the amount of solution ingested by the total number of bees that survived, the consumption was computed. The bees in the CTRL group consumed an average of 0.06 mL of sugar solution each. The DMT, FLU 50, and FLU 100 experimental groups consumed a statistically significant lower amount of solution than the CTRL group. The FLU 100 group consumed an even lower statistically significant (*p* < 0.001) amount of solution than the FLU 50 group.

Regarding pesticide combinations, food consumption was not statistically significantly (*p* > 0.05) different when comparing the DTM group in association with FLU 50 and DMT alone. The difference was, however, statistically significant (*p* < 0.05) if the DTM group in association with FLU 50 was compared with the group administered FLU 50 alone. Consumption of the combination of DTM and FLU 100 was lower and statistically significant (*p* < 0.001) compared to the group receiving DMT alone. However, there was no statistically significant difference (*p* > 0.05) when comparing the DTM and FLU 100 association group with the FLU 100-only group (Figure 2).

### 3.3. Observed Behavior

Figure 3 below represents the percentage of anomalous behaviors recorded in each experimental group.

No anomalous behaviors were recorded in the CTRL group, which were instead present in all the experimental groups treated with the pesticides. The DMT, FLU 50, and FLU 100 groups had a statistically significant (*p* < 0.001) high percentage of abnormal behavior compared to the CTRL group. The FLU 100 group showed an even more pronounced increase in abnormal behaviors than the FLU 50 group (*p* < 0.001). The group in which DMT was associated with FLU 50 had a statistically significantly lower percentage of abnormal behaviors when compared to the DMT group (*p* < 0.05) but significantly higher when compared to the FLU 50 group (*p* < 0.001). When compared to the DMT-alone and FLU 100-alone groups, the DMT and FLU 100 combination did not result in a statistically significant change (*p* > 0.05) in the rate of abnormal behavior (Figure 3).

## 4. Discussion

The crucial role that bees and other pollinating insects play in the pollination of plants has been brought to attention by the reduction in wild pollinators and, in particular, the mortality of domestic bees that has been documented in recent years [39,40,41]. It is difficult to determine the causes of these occurrences since the contributing components can change and interact with each other. To varying degrees, a number of factors, including pesticide exposure, diseases, parasites, beekeeping techniques, and dietary, agro-environmental, and climatic factors, can lead to the weakening and eventual collapse of hives [42,43,44]. As suggested by EFSA [45] and brought up by European Parliament members during European Bee Week 2016 [46], the multifaceted nature of the bee loss issue necessitates a variety of efforts and actions by all parties involved, particularly by scientists who are called upon to comprehend the causes. Pesticides, especially insecticides, are suspected of lowering immune defenses and altering bee behavior, orientation, and social activity when absorbed in sublethal doses. In cases of misapplication (interventions made during flowering, during honeydew flows, in the presence of wind, contamination of spontaneous flora, etc.), pesticides are responsible for extensive mortality [47,48,49]. All these factors are particularly fearsome for the health and survival of pollinators. Honeybees are intimately linked to the environment around the hive, where they come into contact with numerous substances. Usually, honeybees move within a radius of one kilometer to find food resources. If necessary, however, they can extend their range of action [50]. Different pesticides are often sprinkled in the environment; they can be picked up by bees and interact with each other in their organism. In addition, it is often manufacturers who combine the active ingredients of pesticides with each other in order to seek an extension of their spectrum of action or to exploit synergisms of action that make it possible to reduce the concentration of the individual active ingredients, thus releasing fewer toxic substances into the environment [51]. Pesticide mixtures then make treated pest populations less prone to the development of drug resistance [51]. Beyond these advantages, it must be said that the result of pesticide interaction is often unknown. The evaluation of toxicity data for individual compounds may underestimate the risk of pesticide mixtures [52]. Toxicity studies in which the interaction between pesticides is tested are therefore important in order to obtain a clearer picture of the changes that lead to the colony collapse phenomenon in honeybees. Combination exposure to chemicals can be harmful through a variety of processes, such as dosage addition, response addition, and interactions. Interactions can result in antagonistic effects, which result in decreased combined toxicity, or synergistic effects, which result in increased toxicity from interactions between substances. Numerous investigations in this context have found synergistic interactions; nevertheless, in certain circumstances, the concentrations useful for achieving synergistic action are orders of magnitude higher than the usual environmental concentrations [53,54].

In this study, the interactions between DMT and FLU were evaluated. Standard test methods look at the acute effects on adult worker bees for hazard characterization based on mortality 48 h after oral or contact exposure (represented as Lethal Dose for 50% of the individuals). If mortality increases throughout the 48 h test, monitoring might prolong up to 96 h [55,56]. In our previous study, we ascertained the toxic dose of DMT that resulted in the death of the experimental sample of subjects within 72 h [35]. We started with the same dose to determine the interaction effects of DMT with two toxic doses of FLU. The lower dose of FLU (50 mg/L) allowed the bees to survive for up to 3 days, with 46% of subjects still alive; the higher dose (100 mg/L), on the other hand, resulted in the mortality of all treated subjects as early as the second day. The combination of DMT and FLU 50 improved survival compared to the group given DMT alone. The same was evident when the survival rates generated by the DMT and FLU 50 combination were compared with those of the group given FLU 50 alone. In contrast, for the DMT and FLU 100 combination, an improvement in survival indices was noted when these were compared with the group intoxicated with FLU 100 alone, whereas no statistically significant improvement was recorded when the survival indices of this combination were compared with those of the group intoxicated with DMT alone. In general, the recorded improvement in survival rates reflects an antagonism between the two pesticides. This phenomenon can be traced back to a desensitization of the neuronal cells that would no longer respond to the action of the pesticides due to induced overstimulation. As a pyrethroid pesticide, DMT interferes with channel kinetics in the nervous system, causing a delay in their closure and, thus, persistent membrane depolarization [57]. FLU is an acetylcholine receptor (nAChR) agonist. It acts on nerve transmission by mimicking and competing with the natural neurotransmitter [58]. Both pesticides, therefore, act at the nervous level. The overstimulation of neuronal cells and the resulting desensitization could be at the root of this antagonism. The organism of the intoxicated honeybee would then be able to metabolize and excrete pesticides, which could lead to improved survival rates. However, this was not the case when the survival rates of the DMT and FLU 100 combination were compared with the survival rates of the group intoxicated with DMT alone. The latter result can be explained by the hormesis phenomenon. This term refers to a biphasic dose response characterized by a beneficial or stimulatory effect at a low dose and an inhibitory or toxic effect at a high dose [59]. At higher doses of the toxicant, the systems in charge of metabolization, detoxification, and excretion may no longer be able to work effectively and, therefore, would not mitigate intoxication. As for solution consumption, the data must be read in relation to survival indices. As can be seen in Figure 2, the consumption of the contaminated solutions is systematically higher in the test groups in which more subjects survived. It was therefore higher in the control group and the groups intoxicated with FLU 50 alone. Figure 3, which illustrates abnormal behavior, can also be read with the same key. Abnormal behavior was lower in the group intoxicated with the lowest dose of FLU and in those in which antagonism occurred than in the other intoxicated groups.

## 5. Conclusions

Pesticides cause systemic effects on honeybees that can affect their response to numerous environmental stressors. Studies are often carried out on the effects of individual pesticides, but there are few data on the effects generated by their interaction. Interaction studies have often shown synergistic or additive toxicity effects. However, as evidenced by the results of this research, antagonism may also occur. Antagonistic phenomena could be exploited in the formulation of less toxic mixtures that can be released into the environment. Understanding the pharmacological interaction between pesticides is therefore important for the implementation of more sustainable agronomic practices.

## Figures and Tables

**Figure 1 animals-14-00851-f001:**
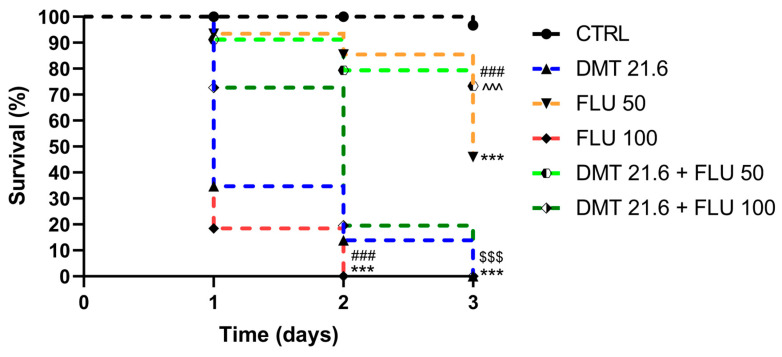
Survival graph. CTRL: control; DMT: deltamethrin; FLU: flupyradifurone. DMT vs. CTRL: *** *p* value < 0.001; FLU 50 vs. CTRL: *** *p* value < 0.001; FLU 100 vs. CTRL: *** *p* value < 0.001; FLU 100 vs. FLU 50: ### *p* value < 0.001; DMT 21.6 + FLU 50 vs. DMT 21.6: ^^^ *p* value < 0.001; DMT 21.6 + FLU 50 vs. FLU 50: ### *p* value < 0.001; DMT 21.6 + FLU 100 vs. DMT 21.6: *p* value > 0.05; DMT 21.6 + FLU 100 vs. FLU 100: $$$ *p* value < 0.001. Ten replicates for each group (total n. = 200 bees per group).

**Figure 2 animals-14-00851-f002:**
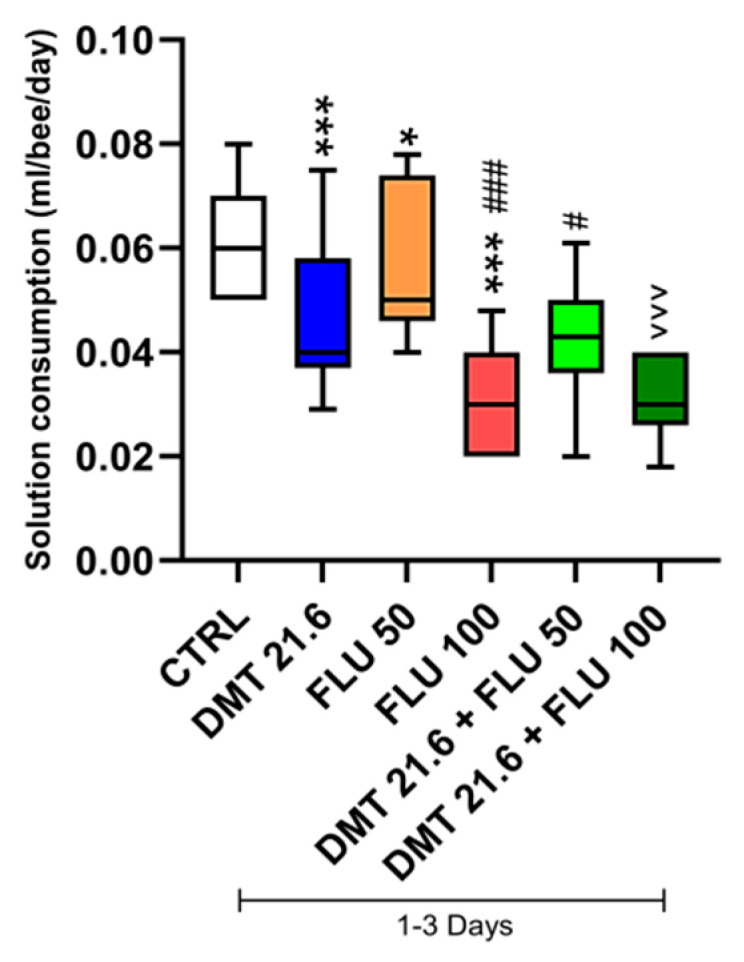
Consumption. CTRL: control; DMT: deltamethrin; FLU: flupyradifurone. * *p* value < 0.05 vs. CTRL; *** *p* value < 0.001 vs. CTRL; # *p* value < 0.05 vs. FLU 50; ### *p* value < 0.001 vs. FLU 50; ^^^ *p* value < 0.001 vs. DMT. The vertical line reports the minimum and maximum values. Ten replicates for each group (total n. = 200 bees per group).

**Figure 3 animals-14-00851-f003:**
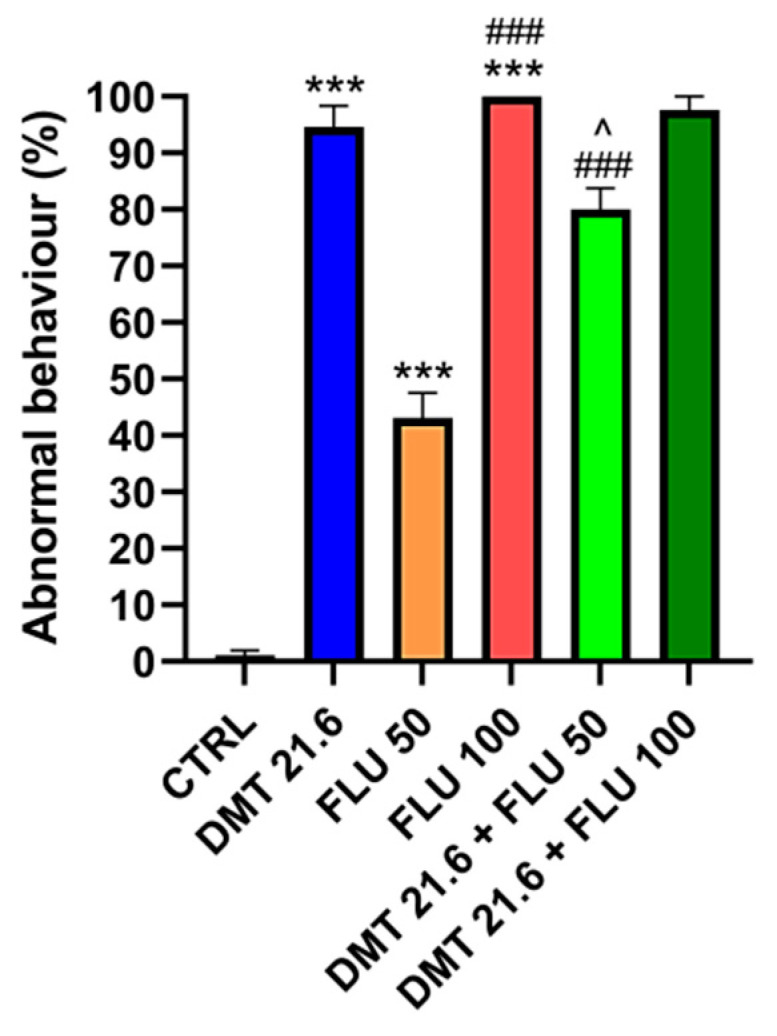
Abnormal behavior. CTRL: control; DMT: deltamethrin; FLU: flupyradifurone. *** *p* value < 0.001 vs. CTRL; ### *p* value < 0.001 vs. FLU 50; ^ *p* value < 0.05 vs. DMT. Data are expressed as mean ± SEM. Ten replicates for each group (total n. = 200 bees per group).

**Table 1 animals-14-00851-t001:** Survival rate in the different groups after 3 days of treatment.

Groups	Survival Rate (%)		
	Day 1	Day 2	Day 3
CTRL	100	100	96.6
DMT 21.6	34.6	13.8	0
FLU 50	93.4	85.4	45.9
FLU 100	18.4	0	/
DMT 21.6 + FLU 50	91.1	79.4	73.3
DMT 21.6 + FLU 100	72.7	19.5	0

## Data Availability

The data are kept at the University of Magna Græcia of Catanzaro and are available upon request.

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
