# Peer review of "Interaction of Flupyradifurone and Deltamethrin, Two Pesticides Commonly Used for Plant Pest Control, in Honeybees"

_animals, 2024, doi:10.3390/ani14060851_

Round 1
Reviewer 1 Report
Comments and Suggestions for Authors
This paper addresses the affects of two pesticides on honey bees, and their interactive effects. This is an important topic, but in some ways the study was not conducted well. The problems are:
1) I am not clear about the number of bees used for each treatment. If there were "ten replicates for each group" (line 136) does that mean 20 x 10 = 200 bees for each treatment? If so, were the survival data used for each treatment pooled for Fig. 1?
2) The observations of abnormal behavior (described in lines 158-162) are very subjective. Other researchers would not be able to repeat this part of the study without some objective guidelines. "Apathy" is a feeling, not a behavior that can be observed. What is "curved belly"? Bees have abdomens, not bellies. If there were 20 bees in a cage, how could the authors observe each bee individually? Bees look a lot alike and are always moving.
3) Fig 2 displays the same data as Fig 1. and should be eliminated.
4) The "hormesis" is interesting but is not explored much. It would be nice to see a series of treatments with cages receiving a wide range of DMT and FLU doses and ratios so that the authors could more accurately determine the doses/ratios at which survival was improved relative to DMT alone. To be convincing, a large sample size is critical here.
Comments on the Quality of English LanguageSome of the English is awkward, but most of the paper can be understood.
Author Response
REVIEWER 1
This paper addresses the effects of two pesticides on honey bees, and their interactive effects. This is an important topic, but in some ways the study was not conducted well.
Response: we deeply thank you for your revision work which helps us improve the quality of the manuscript. We followed your instructions and modified what you suggested us. Changes are highlighted in the manuscript.
The problems are:
1) I am not clear about the number of bees used for each treatment. If there were "ten replicates for each group" (line 136) does that mean 20 x 10 = 200 bees for each treatment? If so, were the survival data used for each treatment pooled for Fig. 1?
R: twenty caged honeybees constituted the experimental unit for each treatment. For each treatment and therefore experimental unit, 10 experimental replicates were set up. So, as you said, 200 honeybees for each treatment. The survival results in Figure 1 are an average of the results obtained from each experimental group. We have now explained this concept better in the manuscript, you will find the changes highlighted.
2) The observations of abnormal behavior (described in lines 158-162) are very subjective. Other researchers would not be able to repeat this part of the study without some objective guidelines. "Apathy" is a feeling, not a behavior that can be observed. What is "curved belly"? Bees have abdomens, not bellies. If there were 20 bees in a cage, how could the authors observe each bee individually? Bees look a lot alike and are always moving.
R: We corrected and replaced belly with abdomen; thanks for the tip. As for behaviors in general, the methods employed and the adjectives used to describe behaviors were not designed by us and we are not the first to have employed them. You can read for example the article by Tosi et al. (2019), https://doi.org/10.1098/rspb.2019.0433, already cited and the OECD guidelines, also cited. Regarding your consideration on the difficulty in observing intoxicated bees individually, Tosi et al. (2019) also observed each honeybee for 6 seconds. At the toxic doses used, the bees are profoundly altered in behavior and do not move as quickly as they do naturally. So, you can clearly observe each bee for its anomalous behaviors.
3) Fig 2 displays the same data as Fig 1. and should be eliminated.
R: We eliminated the figure as you suggested
4) The "hormesis" is interesting but is not explored much. It would be nice to see a series of treatments with cages receiving a wide range of DMT and FLU doses and ratios so that the authors could more accurately determine the doses/ratios at which survival was improved relative to DMT alone. To be convincing, a large sample size is critical here.
R: Your observation is important and we agree with you. As we specified in your first question, the number of experimental replicates was 10 and the survival results were an average of the 10 replicates. Hormesis is a little explored phenomenon and we agree with you that it needs to be taken into greater consideration so that less toxic doses of pesticides can be found in mixtures. This is why we think that studies like the one we conducted are important and should be carried out by multiple research groups.
Some of the English is awkward, but most of the paper can be understood.
R: the text was reviewed and corrected by a native speaker.

Reviewer 2 Report
Comments and Suggestions for Authors
The effects of combined use of two pesticides over different periods of time are of significant scientific and practical interest. The authors of this article use a well-studied model object that is important for agriculture. However, the results obtained by the authors will be interesting for a deeper understanding of the phenomenon of synergy for xenobiotics in general.
Despite this, this article contains several serious shortcomings that do not allow us to recommend it for publication at this time, as well as several less significant technical inconsistencies with international scientific standards.
1. I recommend shortening the title of the article, for example, by removing the words drug, honeybees, experimentally intoxicated by two pesticides commonly used for plant pest control. This will significantly increase the likelihood of other authors citing this article.
2. Lines 25-36 should be halved, removing secondary information. Conversely, the results (lines 36-41) need to be expanded threefold. The abstract largely determines whether readers will want to read the entire article or whether they will leave the article unread.
3. The authors do not indicate on lines 164-166 what method of comparison of samples was used, was there a check for the normality of the sample distribution?
4. Did the authors apply a correction for multiple comparisons of samples? Based on this amendment, the authors' conclusions are unreliable.
5. What does the vertical line in Figures 3, 4 mean (is it an error, standard deviation or standard error)? This must be indicated on lines 164-166 and in the title of each figure.
6. The text of the Results repeatedly states that the reliability of the differences is less than a certain level. This is what they wrote in the 20th century. In the 21st century, it is recommended to use a specific difference significance value and often use a specific Fisher test value to confirm this.
7. Figures 1 and 2 duplicate the same data. One of them needs to be removed.
8. Lines 182-204 are uninformative. We need to rewrite this information more concisely. The same applies to most other parts of the Results section.
9. The title of Figures 3 and 4 should indicate the repetition of experiments.
10. Figure 4 is best depicted as a box analysis (similar to Figure 3).
11. Figures 3 and 4 depict a typical two-factor experiment: FLU (0, 50, 100) and DMT (0, 21.6). These data should definitely be subjected to two-way analysis of variance. This method of analysis will comply with modern scientific standards.
12. The Conclusion section (lines 318-328) cannot contain references to the literature. These proposals should be moved to Discussion.
13. In the bibliography, many sources do not contain all the necessary information (for example, lines 343, 345, 352, 365, 367, 404, 442).
Author Response
REVIEWER 2
The effects of combined use of two pesticides over different periods of time are of significant scientific and practical interest. The authors of this article use a well-studied model object that is important for agriculture. However, the results obtained by the authors will be interesting for a deeper understanding of the phenomenon of synergy for xenobiotics in general.
Despite this, this article contains several serious shortcomings that do not allow us to recommend it for publication at this time, as well as several less significant technical inconsistencies with international scientific standards.
Response: we have answered each of your questions point by point. the text has been particularly revised in the statistical part regarding which you have made several comments. please reread the now modified text, we are sure that your doubts about the scientific consistency of the manuscript can be resolved.
- I recommend shortening the title of the article, for example, by removing the words drug, honeybees, experimentally intoxicated by two pesticides commonly used for plant pest control. This will significantly increase the likelihood of other authors citing this article.
R: thank you for your valuable advice, we have shortened the title as you suggested
- Lines 25-36 should be halved, removing secondary information. Conversely, the results (lines 36-41) need to be expanded threefold. The abstract largely determines whether readers will want to read the entire article or whether they will leave the article unread.
R: thank you for your important advice, the abstract has been modified as per your instructions.
- The authors do not indicate on lines 164-166 what method of comparison of samples was used, was there a check for the normality of the sample distribution?
R: We have added the information.
- Did the authors apply a correction for multiple comparisons of samples? Based on this amendment, the authors' conclusions are unreliable.
R: We have added the information.
- What does the vertical line in Figures 3, 4 mean (is it an error, standard deviation or standard error)? This must be indicated on lines 164-166 and in the title of each figure.
R: We have added the information.
- The text of the Results repeatedly states that the reliability of the differences is less than a certain level. This is what they wrote in the 20th century. In the 21st century, it is recommended to use a specific difference significance value and often use a specific Fisher test value to confirm this.
R: All p values were obtained as a result of a multiple comparison test. Often when this value is so low, the analysis software itself reports the result as p < 0.0001 without specifying the exact number. So even if we reported the exact values, many of them would remain as they are already written.
- Figures 1 and 2 duplicate the same data. One of them needs to be removed.
R: We have deleted figure 2
- Lines 182-204 are uninformative. We need to rewrite this information more concisely. The same applies to most other parts of the Results section.
R: We have arranged the results better.
- The title of Figures 3 and 4 should indicate the repetition of experiments.
R: We have added the information.
- Figure 4 is best depicted as a box analysis (similar to Figure 3).
R: The vertical lines in the box plot do not represent errors but only the maximum and minimum value obtained by an individual bee. In the case of Figure 3 (now become Figure 2), the box plot better represents the data because it shows the range of the lowest and highest value of mL that the bees drank. But in the Figure 4, since these are percentuals of abnormal behaviors, to analyze them, they are reported in the software as zero (to indicate the absence of abnormal behaviors) and 100 (the presence of abnormal behaviors). It follows that via box plot, here the vertical lines would be shown all the way up the scale from zero to 100, because in each group there were bees with or without abnormal behaviors, so graphically it would not be well represented and would lead the reader to think that there is only a very high level of standard error.
- Figures 3 and 4 depict a typical two-factor experiment: FLU (0, 50, 100) and DMT (0, 21.6). These data should definitely be subjected to two-way analysis of variance. This method of analysis will comply with modern scientific standards.
R: To be able to do a two-way ANOVA analysis requires that the data satisfy some important assumptions. One of these is to verify that they have a Gaussian trend and thus pass a parametric test. In our case, the experimental groups did not pass this test. So analyzing them by a two-way ANOVA test would not be correct; nor could a non-parametric two-way ANOVA test be done, since the analysis software does not present this assumption.
- The Conclusion section (lines 318-328) cannot contain references to the literature. These proposals should be moved to Discussion.
R: the sentence with the bibliographical references was moved to the discussions as you suggested.
- In the bibliography, many sources do not contain all the necessary information (for example, lines 343, 345, 352, 365, 367, 404, 442).
R: thank you for your suggestion. The bibliographic references are now modified by adding the missing information.

Reviewer 3 Report
Comments and Suggestions for Authors
The article titled " Flupyradifurone and deltamethrin: drug interaction in honeybees (Apis mellifera) experimentally intoxicated by two pesticides commonly used for plant pest control" investigates the impact of flupyradifurone and deltamethrin on honey bees, particularly focusing on survival, sucrose intake and abnormal behaviors.
1. The authors used basic research methods to examine the impact of two pesticides on bees, showing limited innovation in their approach. Additionally, the concentrations of the pesticides applied in the experiments were substantially higher than the actual residual concentrations encountered in the field. Employing such elevated concentrations to evaluate their effects on bee health seems unjustifiable.
2. The method used by the authors to evaluate abnormal behavior is too vague, making it difficult to define and measure. Even though there is some support from the literature, I think it's not rigorous enough. If it's necessary to assess abnormal behavior, I suggest using methods that are easier to quantify.
3. For the results in Figure 3, FLU 100 group have all dead in day 2. How could you used the solution consumption amount of 1-3 days to plot the figure? For me, the FLU concentration of 50 and 100 are too high, which led to too high mortality.
4. It is better to mark the significance in Figure 1 and 2.
Author Response
REVIEWER 3
The article titled " Flupyradifurone and deltamethrin: drug interaction in honeybees (Apis mellifera) experimentally intoxicated by two pesticides commonly used for plant pest control" investigates the impact of flupyradifurone and deltamethrin on honey bees, particularly focusing on survival, sucrose intake and abnormal behaviors.
Response: we thank you for your important review work which helps us improve the overall quality of the manuscript. We have responded point by point to your comments, editing where necessary in the text.
- The authors used basic research methods to examine the impact of two pesticides on bees, showing limited innovation in their approach. Additionally, the concentrations of the pesticides applied in the experiments were substantially higher than the actual residual concentrations encountered in the field. Employing such elevated concentrations to evaluate their effects on bee health seems unjustifiable.
R: the methods used have also been used by other research groups; you can read for example the article by Tosi et al. (2019), https://doi.org/10.1098/rspb.2019.0433. The toxic concentrations used are not field doses but, in any case, are toxic doses that have recently been used by other researchers for acute toxicity studies. Please refer to the following studies: https://doi.org/10.3390/ani13243764, https://doi.org/10.3390/ani14040608, https://doi.org/10.3389/fphys.2023.1150340, https://doi.org/10.1002/etc.67. These doses allowed us to obtain a mortality rate in three days, therefore due to acute intoxication.
- The method used by the authors to evaluate abnormal behavior is too vague, making it difficult to define and measure. Even though there is some support from the literature, I think it's not rigorous enough. If it's necessary to assess abnormal behavior, I suggest using methods that are easier to quantify.
R: as you stated, the methods used have support from the literature. In particular, we referred to works such as that of Tosi et al. (2019), https://doi.org/10.1098/rspb.2019.0433. We believe this is quite sufficient to consider the results obtained as scientifically valid. However, we will treasure his advice as we try to devise more accurate methods in our subsequent experiments.
- For the results in Figure 3, FLU 100 group have all dead in day 2. How could you used the solution consumption amount of 1-3 days to plot the figure? For me, the FLU concentration of 50 and 100 are too high, which led to too high mortality.
R: we understand your comment, but we reiterate that we used doses capable of causing acute toxicity. We were not the first to use the doses used to test the toxicity of deltamethrin and flupyradifurone, please refer to the following studies: https://doi.org/10.3390/ani13243764, https://doi.org/10.3390/ani14040608, https://doi.org/10.3389/fphys.2023.1150340, https://doi.org/10.1002/etc.67.
- It is better to mark the significance in Figure 1 and 2.
R: we have added this information to the figures

Reviewer 4 Report
Comments and Suggestions for Authors
The present paper describes interaction flupyradifurone (FLU) and deltamethrin (DMT) on honeybees.
The main purpose is to describe the mortality and behavior of honeybee contaminated through alimentation by FLU and DMT alone or in combination.
The subject seems to be promising, particularly to give a better understanding of the interactions between pesticides.
The manuscript can be accepted with minor revision due to lack of information and approximations of the text.
1. Introduction
Lines 47-48: “Their use … for society” is too imprecise.
Lines 53-54: “The flupyradifurone … market” should be associated to the second paragraph instead of the first one.
Lines 62-76: I think the description of general antagonist/synergist effects should be replaced by a description of the deltamethrin effects and use. The join use of the two pesticides (Bayer patent?) must be mentioned: formulation of the two pesticides, joint presence on crops?
2. Materials and methods
Lines 137-138: are the pesticides doses environmental realistic? How are they chosen?
Line 140: What do you intend with “a toxic dose of DTM”? Mortality?
3. Results
Line 169: “percentageof” should be replaced by “percentage of”
4. Discussion
Lines 255-256: Honeybee fly 3km around the hive to find resources.
Line 269: “interactions Interactions” should be replaced by “interactions. Interactions”
Lines 279-282: Information regarding concentration choice should be placed earlier in the text to have a better understanding of the experimental design.
Author Response
REVIEWER 4
The present paper describes interaction flupyradifurone (FLU) and deltamethrin (DMT) on honeybees.
The main purpose is to describe the mortality and behavior of honeybee contaminated through alimentation by FLU and DMT alone or in combination.
The subject seems to be promising, particularly to give a better understanding of the interactions between pesticides.
The manuscript can be accepted with minor revision due to lack of information and approximations of the text.
Response: we thank you deeply for your review work and for your appreciation of the topic covered. We followed your advice and made the changes to the text. You will find the changes highlighted.
- Introduction
Lines 47-48: “Their use … for society” is too imprecise.
R: thanks for the report, the sentence has been reformulated
Lines 53-54: “The flupyradifurone … market” should be associated to the second paragraph instead of the first one.
R: we moved the sentence to the beginning of the second paragraph as you suggested
Lines 62-76: I think the description of general antagonist/synergist effects should be replaced by a description of the deltamethrin effects and use. The join use of the two pesticides (Bayer patent?) must be mentioned: formulation of the two pesticides, joint presence on crops?
R: we thank you for your important advice to help us improve the quality of the manuscript. We have added the information about the toxic effects of deltamethrin. However, with your consent we would prefer to keep the information about drug interactions of synergism and antagonism. We believe they are important as they are the cornerstone on which the basis and results of the article are based and also result in the keywords.
- Materials and methods
Lines 137-138: are the pesticides doses environmental realistic? How are they chosen?
R: The toxic doses were chosen in relation to information that you can find in other previously published studies; two of these studies were recently conducted by our research group. Specifically, the following are the doi of the papers we refer to: https://doi.org/10.3390/ani13243764, https://doi.org/10.3390/ani14040608, https://doi.org/10.3389/fphys.2023.1150340, https://doi.org/10.1002/etc.67. These doses allowed us to obtain a mortality rate in three days, therefore due to acute intoxication.
Line 140: What do you intend with “a toxic dose of DTM”? Mortality?
R: this dose was the toxic one obtained from previous studies and specifically https://doi.org/10.1002/etc.67 and https://doi.org/10.3390/ani13243764. This dose allowed us to obtain a mortality rate in three days, therefore due to acute intoxication.
- Results
Line 169: “percentageof” should be replaced by “percentage of”
R: now amended
- Discussion
Lines 255-256: Honeybee fly 3km around the hive to find resources.
R: Thanks for this comment. We have added the information in the discussion section also adding a bibliographical note
Line 269: “interactions Interactions” should be replaced by “interactions. Interactions”
R: Now amended
Lines 279-282: Information regarding concentration choice should be placed earlier in the text to have a better understanding of the experimental design.
R: We have specified this information in the text in section 2.2. Pesticide concentration and administration

Round 2
Reviewer 1 Report
Comments and Suggestions for Authors
I accept the proposed revisions. But I am still opposed to the term "apathy", line 184. This is a feeling, not an observable trait, and the authors do not know the feelings of the bees.
Author Response
I accept the proposed revisions. But I am still opposed to the term "apathy", line 184. This is a feeling, not an observable trait, and the authors do not know the feelings of the bees.
Response: we deeply thank the reviewer for his important revision work that helped us to improve the quality of the manuscript. We have replaced the term "apathy" with the expression "decrease in movement activity". This is what we meant by the term apathy.
Reviewer 3 Report
Comments and Suggestions for Authors
I do not have any more questions.
Author Response
I do not have any more questions.
Response: we deeply thank the reviewer for his important revision work that helped us improve the quality of the manuscript.